# Mechanical Properties of Macromolecular Separators for Lithium-Ion Batteries Based on Nanoindentation Experiment

**DOI:** 10.3390/polym14173664

**Published:** 2022-09-03

**Authors:** Wenqian Hao, Xiqiao Bo, Jiamiao Xie, Tingting Xu

**Affiliations:** 1The College of Mechatronic Engineering, North University of China, Taiyuan 030051, China; 2Underground Target Damage Technology National Defense Key Discipline Laboratory, North University of China, Taiyuan 030051, China; 3Guangdong Aerospace Research Academy, Guangzhou 511466, China; 4School of Chemistry and Chemical Engineering, Northwestern Polytechnical University, Xi’an 710129, China

**Keywords:** macromolecular separator, lithium-ion batteries, hardness, elastic modulus, nanoindentation

## Abstract

High tensile strength and toughness play an important role in improving the mechanical performance of separator films, such as resistance to external force, improving service life, etc. In this study, a nanoindentation experiment is performed to investigate the mechanical properties of two types of separators for LIBs based on the grid nanoindentation method. During the indentation experiment, the “sink-in” phenomenon is observed around the indenter when plastic deformation of the specimen occurs. The “sink-in” area of the polyethylene (PE) separator is larger than that of the polypropylene/polyethylene/polypropylene (PP/PE/PP) separator, i.e., the plastic area of the PE separator is larger than that of the PP/PE/PP separator. In order to select a suitable method to evaluate the hardness and elastic modulus of these separators for LIBs, three theoretical methods, including the Oliver–Pharr method, the indentation work method, and the fitting curve method, are used for analysis and comparison in this study. The results obtained by the fitting curve method are more reasonable and accurate, which not only avoids the problem of the large contact area obtained by the Oliver–Pharr method, but also avoids the influence caused by the large fitting data of the displacement–force curve and the inaccuracy of using the maximum displacement obtained by the indentation method. In addition, the obstruction ability of the PP/PE/PP separator to locally resist external load pressed into its surface and to resist micro particles, such as fine metal powder, that can enter the lithium-ion battery during the manufacturing process is greater than that of the PE separator. This research provides guidance for studying the mechanical properties and exploring the estimation method of macromolecular separators for LIBs.

## 1. Introduction

Based on the advantages of high energy density, long cycle life, zero memory effect, low self-discharge, and environmental friendliness, lithium-ion batteries (LIBs) have been widely used in electrochemical energy conversion and storage, including in cell phones, electric vehicles, etc. [1,2,3,4]. LIBs mainly consist of a positive electrode, negative electrode, electrolytes, and macromolecular separator. The separator is set up between positive and negative electrodes to prevent internal short circuiting of the LIBs [5,6]. The lithium dendrites may puncture the separator when it is compressed by external force, which can lead to internal short circuit or explosion of LIBs due to the direct contact between positive and negative electrodes [3,5]. According to the requirements of the United States Advanced Battery Consortium (USABC) for lithium-ion battery separators, the specifications of separators immersed in liquid electrolyte are >300 g/25.4 μm puncture strength and <2% offset at 1000 psi tensile strength. Generally, the mechanical properties of positive and negative electrodes determine the hardness and elastic modulus of separators. There are two kinds of internal short circuits possible in LIBs because the conductive particles exist between the positive and negative electrodes of the battery. Accordingly, the material properties of electrodes determine the ability of the separator to prevent internal short circuits, which depends on the battery’s form. The electrodes of the LIBs experience successive expansions and contractions during charge and discharge cycling. For pouch batteries, trapped conducting particles experience a periodically varying force perpendicular to the plane of the electrodes and will produce an internal short circuit after piercing through the separator. Therefore, the puncture resistance of the separator is a key functionality concern. For cylindrical batteries, successive expansions and contractions of the electrodes cause a periodic tangential motion of the electrodes during prolonged charge–discharge cycling. The conductive particles between the positive and negative electrodes will perform a see-saw motion and will cause the internal short circuit after cutting through the separator. Therefore, the abrasion resistance of separator is a key functionality concern. Therefore, high tensile strength and hardness play an important role in improving the mechanical properties of the macromolecular separator, such as resistance to external force, improving service life, and so on. The present work aims to quantify the puncture resistance of separator by measuring the hardness and elastic modulus through nanoindentation experiments.

Many researchers strove to develop the electrochemical performance [7,8,9,10,11] and mechanical properties [12,13,14,15,16] of LIBs, such as the stress distribution of separators during charge and discharge processes [10], the influence of separator creep on battery capacity [15], and the relationship between separator strain and lithium-ion transport [16]. However, the separator was regarded as a homogeneous material in previous studies, the research variables were the average mechanical properties and the experiment result was the macroscopic average result [10,11,14]. In order to study the influence of the electrode’s active particles, the metal particle powder during the fabrication, and the lithiation properties during charge and discharge on the mechanical and electrochemical performances of LIBs, the microscopic mechanical properties of the separator need to be studied and the hardness and elastic modulus need to be determined. The importance of the effects of micromechanical properties on separators is indeed the motivation of recent studies because damage to the separator is largely caused by microscale particles.

The penetration resistance of separator can be visually illustrated by the hardness and elastic modulus because the hardness and elastic modulus characterize the resistance ability of permanent deformation and the elastic deformation, respectively. Generally, the hardness and elastic modulus are determined by nanoindentation experiment from the micro point of view [17]. The elastic modulus of separator is determined by the unloading part of the loading–unloading cycle curve, and the hardness is calculated from the residual indentation area of the separator after its plastic deformation exceeds its elastic limit. The grid nanoindentation method is one of the most valuable applications in all kinds of microporous or foam material nanoindentation experiments [18,19,20]. The grid nanoindentation experiment is a regular continuous experiment determined by the relative size of nanoindentation and microstructure characteristic of materials. This experiment reflects the effective mechanical performance of the microporous material, determines the local distribution of hardness and elastic modulus around the nanoregion and characterizes uniformly composite or heterogeneous components with corresponding properties.

During the indentation experiment, the stacking effect or “sink-in” phenomenon was observed around the indenter when plastic deformation of the specimen occurs [21,22]. This assumption could be used to calculate the hardness and elastic modulus of the specimen based on the displacement–force curve to obtain the projected contact area [23]. For the sharp indentation experiment, the material may sink in and stack around the indenter when plastic deformation occurs. The material stacking causes the change of actual contact area between the indenter and specimen, which leads to the estimation errors due to contact areas. The stacking effect is observed around the indenter tip for the soft material, which overestimates the hardness and elastic modulus of specimen due to the increased contact area between the indenter and specimen. The indentation effect is observed around the indenter tip for the hard material, which underestimates the hardness and elastic modulus of specimen due to the decreased contact area between the indenter and specimen [24,25]. The contact area may be underestimated by about 60% when the stacking effect occurs, based on the finite element study conducted by Bolshakov et al. [26]. These results depended on the ratio of the reduced modulus to yield stress and the work hardening properties of the material. For indentation experiments of non-uniform materials, the influence of the stacking effect aggravated. The constraints were imposed by the fibers around the fiber composite on the subsurface stress field conducted by Hardiman et al. [27]. The results showed that the matrix components may produce the stacking effect in the process of the indentation experiment. In addition, Tranchida et al. [28] and Hardiman et al. [29] used scanning probe microscopy to study the stacking effect on the indentation property of polymer from the nanoscale and microscopic scale, respectively. The results showed that the overestimation of indentation modulus of polymer may be due to the stacking effect.

The mechanical parameters (hardness and elastic modulus) are determined by different methods, including Oliver–Pharr method, indentation work method, and fitting curve method. However, there is relatively little research about the comparison of the contact area based on different quantitative evaluation methods in recent years. In addition, there is little nanoindentation research on macromolecular separators used in LIBs. Therefore, the nanoindentation experiment is performed to investigate the mechanical properties of two separator types for LIBs based on the grid nanoindentation method. In order to determine a suitable method to evaluate the hardness and elastic modulus of these separators for LIBs, three theoretical methods, including the Oliver–Pharr method, the indentation work method, and the fitting curve method, are used for analysis and comparison in this study. This research provides guidance for studying the mechanical properties and exploring the estimation method of macromolecular separators for LIBs.

## 2. Materials and Methods

In order to investigate mechanical properties of separators for LIBs based on nanoindentation experiment, three typical macromolecular separators were selected, the first was a tri-layer polypropylene/polyethylene/polypropylene (PP/PE/PP) separator (thickness: 25 µm; porosity: 39%) prepared by dry process and the second was a single-layer polyethylene (PE) separator (thickness: 25 µm; porosity: 40%) prepared by wet process. The vendor of PP/PE/PP and PE separators used in this work is Celgard company. The thickness, permeability, porosity, PP pore size, and shrinkage of the separator are important parameters which characterize the stability and long cycle life of LIBs. The fundamental properties of the separators are summarized in Table 1 [30,31,32]. It is noted that the permeability of separators can be described by the MacMullin number. The MacMullin number is proportional to air permeability and it is often expressed by the JIS (Japanese Industrial Standards) Gurley value [30]. As a functional material, there was an obvious anisotropic effect of the separator, due to its manufacturing procedure (submicron pore) and mechanical properties (tensile strength) difference between machine direction (MD) and transverse direction (TD) [33]. The pore size should be small enough to prevent the penetration of the electrode particles and conducting additives. The uniform distribution and exacting structure of the pore size contribute to the inhibition of lithium dendrites. Typically, a pore size of <1 µm is desirable for separators used in LIBs. The porosity of the separator is defined by the weights of separator before and after liquid absorption, i.e., porosity = [(Ω − Ω_0_)/*ρV*_0_] × 100%, where Ω_0_ and Ω are the weights of separator before and after immersing the liquid, respectively, *ρ* is the density of the liquid, and *V*_0_ is the geometric volume of the separator [34]. The desirable porosity of normal LIBs separator is about 40%~60%. The thermal shrinkage (<5%) at 90 °C/1 h is also determined by orthogonal directions.

The microstructures of in-plane views of separators were observed using an extrahigh-resolution field emission scanning electron microscope (FE-SEM) FEI Verios G4 (FEI Company, Hillsboro, OR, USA) in a vacuum environment with 5 kV accelerating voltages, as shown in Figure 1. All separators were sputter-coated with Au-Pd alloy using a magnetron ion sputtering instrument to improve image quality because these polymer separators are non-conductive during FE-SEM measurement. It can be found that the micromorphology of the three specimens was distinctive due to the different preparation processes of dry and wet methods. The micropores of PP/PE/PP separator (specimen 1) were elliptic with a clear orientation parallel along the deformation direction and the pore size was about 0.5 μm × 0.05 μm. In addition, Halalay [35] has obtained a cross section electron microscope of the tri-layer PP/PE/PP separator and pointed out that the thickness of three layers of different materials was the same. Unlike specimen 1, the micropores of the PE separator (specimen 2), shown in Figure 1c,d, had a pore size less than 1 μm.

The actual working environment of LIBs is between positive and negative electrodes. The average size of the embedded stress for micro particles is about 0.5 μm~1.0 μm along the direction of separator thickness [10]. In addition, the specimen thickness should be large enough, or indentation depth small enough, that the experimental result is not influenced by the specimen support. According to the American Society for Testing and Materials standards (ASTM E2546-15) [36], the specimen thickness was greater than or equal to 10 times of the indentation depth or 6 times of the indentation radius. Therefore, each specimen was tested based on displacement control method in the process of the experiment, that is, the indentation depth was set to 0.5 μm, which basically equaled to the micropore size of the separators. For the indentation spacing adopted by the grid indentation method, the influence of the plastic deformation caused by different indentation should be taken as the criterion. According to the ASTM E2546-15 [36], the indentation spacing between adjacent indentation points was at least 10 times that of the indentation radius, so that the indentation spacing is determined to be 10 μm. The PP thickness on the first layer of specimen 1 is about 8 μm, which far more than 10 times the indentation depth of 0.5 μm. Thus, the influence of the second PE layer on the substrate of specimen 1 in the nanoindentation experiment at the indentation depth of 0.5 μm can be ignored. The two tested specimens were thin and soft, so the substrate needed to be very flat due to the micropores existing in the separator. To ensure the consistency of specimen geometry, an acrylic template with 0.5 mm thickness was milled. Specimens were cut along the perimeter of the acrylic template on transparent Cartesian graph paper to improve dimensional accuracy and cut quality. The microscope slide of 25.4 mm × 38.1 mm × 1.0 mm was selected as the substrate in the nanoindentation experiment. Because a large amount of released heat during the bonding process led to the modification of separator, so ethyl α-cyanoacrylate glue was not suitable for bonding the separator and substrate. Therefore, the sodium carboxymethyl cellulose was used as an adhesive for the separator and substrate. The viscosity of sodium carboxymethyl cellulose can be controlled by adjusting the water consumption.

The nanoindentation experiment was performed by Ultra nano indentation tester Anton Paar UNHT^3^ (Anton Paar Group, Ashland, VA, USA), which was suitable for the nano-scale measurement of polymers. The indenter with Berkovich tip was used and the length of the base side of indenter was 50 μm. The ratio of the height to the length between the tip to the bottom side was 1:7. The influence of the indentation radius R of the indenter tip on the hardness experiment results could be ignored when the indentation depth was close to 0.5 μm. The temperature was set to 25 °C and the relative humidity was set from 25% to 30%.

Before the nanoindentation experiment, the area calibration was carried out by a pre-loading experiment for each specimen to determine the friction and initial contact point between the indenter and specimen. After the pre-compression experiment, the instrument remained stable, and nanoindentation experiment was performed on the 3 × 3 matrix points with indentation depth of 0.5 μm and indentation spacing of 10 μm. The control parameters of the nanoindentation experiment are listed in Table 2. The indentation loading was kept for 5 s when the maximum displacement was reached. The loading stage of the nanoindentation experiment was divided into three stages: loading stage, loading retention stage, and unloading stage, as shown in Figure 2.

## 3. Results and Discussion

### 3.1. Nanoindentation Experiment Results

Nine indentation points were selected to obtain the displacement–force curves of specimen 1 and specimen 2, as shown in Figure 3. All the displacement–force curves show the hysteresis and creep characteristics, which are typical viscoelastic-plastic polymer behavior. The displacement–force curve is divided into two phases, the loading curve phase and unloading curve phase. The loading curve phase is the force generated when the indenter is initially pressed into the specimen. Due to the initial elasticity of the specimen, the unloading curve phase is the force generated after the specimen springs back. The inflection point (maximum indentation depth) corresponds to the maximum force on the specimens. The displacement of unloading curve phase corresponds to the final indentation depth. The maximum force of specimen 1 is between 0.19 mN and 0.21 mN, and the maximum force of specimen 2 is between 0.11 mN and 0.16 mN. Under the same displacement, the forces of specimens in different indentation points are different because the microstructures of specimens are different in different indentation positions. The consistency exists in the displacement–force curves of specimen 1 because the PP/PE/PP separator structure is more uniform (Figure 1a,b), as shown in Figure 3a. The distribution range and fluctuation degree of displacement–force curves of specimen 2 are large because the nonuniform micropores and unequal ribs diameter existed in the PE separator structure (Figure 1c,d), as shown in Figure 3b. The total work *W_t_* can be obtained by the area below the displacement–force curve, and elastic work *W_e_* can be obtained by the area below the unloading curve. The difference between the total work and the elastic work is the work done by the plastic deformation *W_p_* of the specimen. According to Figure 3, the total work, elastic work, and plastic work of specimens at different indentation points can be determined, as shown in Section 3.5. In addition, maximum indentation depth and final indentation depth after nanoindentation experiment also can be determined, as shown in Section 3.5. After the nanoindentation experiment, the microstructures of in-plane views of two types specimens are observed using an extra-high resolution field emission scanning electron microscope FEI Verios G4 (FEI Company, Hillsboro, Oregon, USA) in a vacuum environment with 3 kV accelerating voltages, as shown in Figure 4. The finial indentation depth in Figure 4 corresponds to the finial displacement of unloading curve phase in Figure 3. During the indentation experiment, the “sink-in” phenomenon is observed around the indenter when plastic deformation of specimen occurs. The “sink-in” area of PE separator is larger than that of PP/PE/PP separator, i.e., the plastic area of the PE separator is larger than that of the PP/PE/PP separator. Thus, the elastic area of the PE separator is smaller than that of the PP/PE/PP separator. According to Figure 4, the indenter is not in full contact with the specimen due to the “sink-in” effect. The actual contact depth between the indenter and the specimen can be expressed as the contact depth measured by micromorphology.

### 3.2. Mechanical Parameters Based on the Oliver–Pharr Method

The mechanical parameters are determined by different methods, including Oliver–Pharr method, indentation work method and fitting curve method. Based on previous research, the method of hardness and elastic modulus of specimen were proposed by Oliver and Pharr [24,25] through calculation of the slope and maximum indentation depth in unloading points of displacement–force curves. Although it was originally intended for application with sharp, geometrically self-similar indenters, such as the Berkovich triangular pyramid, Oliver and Pharr have since realized that it is much more general than this and applies to a variety of axisymmetric indenter geometries, including the sphere [37]. The schematic illustration of theoretical model of the Oliver–Pharr method is shown in Figure 5, where the parameter *P* designates the force and *h* the displacement relative to the initial undeformed surface. Some important quantities must be determined from the displacement–force curve, including the maximum force *P*_max_, the maximum displacement *h*_max_, the final indentation depth after unloading *h_f_*, and the elastic unloading stiffness, *S* = d*P*/d*h*, defined as the slope of the upper portion of the unloading curve during the initial stages of unloading (also called the contact stiffness). According to Oliver and Pharr’s theory [24,25], the unloading curves are expressed by the power law relation (Figure 5a) as
(1)$$P=\alpha {\left(h-h_{f}\right)}^{m}$$
where *α* and *m* are power fitting constants. When the indenter is a flat punch, constant is *m* = 1; when the indenter is Berkovich cone, constant is *m* = 2; and when the indenter is a spherical punch, constant is *m* = 1.5 [37,38,39]. The slope of the upper portion of the unloading curve during the initial stages of unloading, i.e., contact stiffness, are given by
(2)$$S=\beta \frac{2}{\sqrt{\pi }}E_{eff}\sqrt{A}$$
where *β* is the constant related to the head geometry, when indenter is Berkovich cone, *β* is 1.034. *A* is the contact area between the indenter and the specimen. *E_eff_* is the effective elastic modulus defined by
(3)$$\frac{1}{E_{eff}}=\frac{1-v^{2}}{E}+\frac{1-v_{i}^{2}}{E_{i}}$$
where *E* and *E_i_* are the elastic modulus of specimen and indenter, respectively. *v* and *v_i_* are the Poisson’s ratio of specimen and indenter, respectively. The material of the indenter is diamond, the elastic modulus and Poisson’s ratio are *E_i_* = 1141 GPa and *v_i_* = 0.07. The effective elastic modulus takes into account the fact that elastic displacements occur in both specimens.

The parameters in the loading and unloading process characterizing the contact geometry are shown in Figure 5b, in which it is assumed that the behavior of the Berkovich indenter can be modeled by a conical indenter with a half-included angle, *ϕ*, that gives the same depth-to-area relationship, *ϕ* = 70.3° [24]. During the indentation experiment, the “sink-in” phenomenon is observed around the indenter when plastic deformation of specimen occurs. The amount of sink-in, *h_s_*, is given by
(4)$$h_{s}=\varepsilon \frac{P_{\max}}{S}$$
where *ε* is a constant that depends on the indenter geometry, *ε =* 0.72 for a conical punch, *ε =* 0.75 for a revolution paraboloid punch and *ε =* 1.00 for a flat punch [40].

The final indentation depth *h_f_* between the indenter and the specimen after unloading is given by
(5)$$h_{c}=h_{\max}-h_{s}=h_{\max}-\varepsilon \frac{P_{\max}}{S}$$

The contact area *A* between the indenter and the specimen is a function of final indentation depth *h_f_*, this area function can be expressed as
(6)$$A=A(h_{c})=\sum_{n=0}^{8}Cn(h_{c}^{2-n})=24.5h_{c}^{2}+C_{1}h_{c}+C_{2}h_{c}^{\frac{1}{2}}+C_{1}h_{c}^{\frac{1}{4}}+\cdot \cdot \cdot \cdot \cdot \cdot +C_{8}h_{c}^{\frac{1}{128}}$$

Once the contact area is determined, the hardness *H* can be determined by
(7)$$H=\frac{P_{\max}}{A}$$

According to Equations (1)–(7), the elastic modulus *E* and hardness *H* can be determined. The hardness and elastic modulus of specimens in different indentation points based on the Oliver–Pharr method are shown in Figure 6. The hardness of PP/PE/PP separator (specimen 1) and PE separator (specimen 2) is between 35.70 MPa and 41.17 MPa, and 16.10 MPa and 25.13 MPa, respectively. The elastic modulus of PP/PE/PP separator (specimen 1) and PE separator (specimen 2) is between 0.62 GPa and 0.69 GPa, and 0.64 GPa and 0.97 GPa, respectively. The hardness of the PP/PE/PP separator (specimen 1) is greater than that of the PE separator (specimen 2), while the elastic modulus of the PP/PE/PP separator (specimen 1) is smaller than that of the PE separator (specimen 2) as a whole when the thickness is same.

### 3.3. Mechanical Parameters Based on the Indentation Work Method

In order to deal with the influence of the stacking effect on the nanoindentation experiment results, Stillwell and Tabor [22] proposed the indentation work method to measure the hardness and elastic modulus of specimens. The indentation work method does not need to calculate the contact area of the indentation, but only uses the dissipated energy and the work done during the indentation process to calculate the material properties, that is, the closed area formed by the displacement–force curve and *x*-axis are calculated. The indentation work method essentially uses the plastic work divided by the indentation volume, as
(8)$$\frac{P}{A_{P}}=\frac{W_{P}}{V_{P}}$$
where *P* is force, *A_p_* is the area of the plastic zone, *V_p_* is the volume of the total plastic deformation, which is the sum of the indentation volume and the stacking effect volume. *W_p_* is the plastic work associated with the toughness of the material, which reflect absorbed energy of material during the indentation process. The total work *W_t_* can be obtained by the area below displacement–force curve, and elastic work *W_e_* can be obtained by the area below unloading curve, as shown in Figure 5a. The plastic work is the difference between the total work and the elastic work [41], expressed as
(9)$$W_{p}=W_{t}-W_{e}$$

The total work *W_t_* can be obtained by the area below displacement–force curve (Figure 5a), which is given by
(10)$$W_{t}=\int_{0}^{h_{\max}}P(h)dh$$

The typical displacement–force response of an elasto-plastic material to a self-similar sharp indentation can be well described by the Kick’s law (*P* = *Ch*^2^) [42,43], where $C=P_{\max}/h_{\max}^{2}$ is the slope of loading curve, which reflects the ability of material to resist plastic deformation caused by external force. *P*_max_ is the maximum force, *h*_max_ is the maximum indentation depth. Substituting the Kick’s law into Equation (10), it can be obtained by
(11)$$W_{t}=\int_{0}^{h_{\max}}Ch^{2}dh=\frac{Ch_{\max}^{3}}{3}=\frac{P_{\max}h_{\max}^{}}{3}$$

It has been shown that the ratio of *h_f_*/*h*_max_ is equivalent to that of *W_p_*/*W_t_* [22,41,44]. Thus, the ratio of elastic work and total work can be expressed by
(12)$$\frac{W_{e}}{W_{t}}=1-\frac{W_{p}}{W_{t}}=1-\frac{h_{f}^{}}{h_{\max}^{}}$$

The indentation work method was suggested by Tuck et al. [45,46,47], who found that the hardness could be calculated on the basis indentation work alone, and can be represented by
(13)$$H=\frac{\kappa P_{\max}^{}}{h_{c}^{2}}=\frac{\kappa P_{\max}^{3}}{9W_{t}^{2}}$$
where *h*_c_ is contact depth, *κ* is a constant equal to 0.0408 for the three-sided Berkovich pyramidal indenter.

In addition, by taking the hardness to be based on plastic deformation alone, the work done should then be the plastic work and the following equation is given by [47]
(14)$$H=\frac{\kappa P_{\max}^{3}}{9W_{p}^{2}}$$

The plastic depth can be calculated by
(15)$$h_{p}=\frac{3W_{p}^{}}{P_{\max}^{}}$$

According to Equations (9)–(14), the hardness of specimens can be calculated. In addition, the elastic modulus can be calculated by
(16)$$\frac{W_{t}-W_{e}}{W_{t}}=1-\frac{5H}{E}$$

The indentation work method is usually used to measure the mechanical properties of soft materials compressed by lower load [47]. The hardness and elastic modulus are obtained by this method to process the load–displacement curve obtained in the experiment, as shown in Figure 7. The hardness of the PP/PE/PP separator (specimen 1) and PE separator (specimen 2) is between 21.92 MPa and 24.78 MPa, and 12.39 MPa and 21.62 MPa, respectively. The elastic modulus of PP/PE/PP separator (specimen 1) and PE separator (specimen 2) is between 0.30 GPa and 0.33 GPa, and 0.34 GPa and 0.48 GPa, respectively. The hardness of the PP/PE/PP separator (specimen 1) is greater than that of the PE separator (specimen 2), while the elastic modulus of the PP/PE/PP separator (specimen 1) is smaller than that of the PE separator (specimen 2) as a whole when the thickness is same.

### 3.4. Mechanical Parameters Based on the Fitting Curve Method

Kick’s law (*P* = *Ch^2^*) and maximum displacement *h*_max_ are adopted in the indentation work method. But the relationship between displacement and force is not always quadratic. In addition, there is the contact depth *h*_c_ between the indenter and specimen, which is smaller than the maximum displacement *h*_max_ because of the “sink-in” phenomenon, as shown in Figure 5b. Therefore, it is assumed that the fitting curve between displacement and force satisfies the expression *P* = *Ch^n^*, where *n* is the fitting exponential. Substituting the expression *P* = *Ch^n^* and $P_{\max}=Ch_{\max}^{n}$ into Equation (10), it can be obtained by
(17)$$W_{t}=\int_{0}^{h_{\max}}Ch^{n}dh=\frac{Ch_{\max}^{n+1}}{n+1}=\frac{P_{\max}h_{\max}}{n+1}$$

According to Equations (13) and (17), the hardness of specimens can be calculated by
(18)$$H=\frac{\kappa \left(n+1\right)W_{t}}{h_{c}^{2}h_{\max}}$$

The relationship between the contact depth and maximum displacement can be calculated by
(19)$$h_{c}=\frac{2(\nu_{e}-1)}{2\nu_{e}-1}h_{\max}$$
where *v_e_* is energy constant, which is defined as:(20)$$\nu_{e}=\frac{W_{s}}{W_{e}}$$
where *W*_s_ = *P*_max_*h*_max_ is the absolute work, which is linearly related to the total work *W_t_*, plastic work *W_p_* and elastic work *W_e_*.

Substituting Equation (19) into Equation (18), the hardness can be obtained by
(21)$$H=\frac{\kappa \left(n+1\right){\left(2\nu_{e}-1\right)}^{2}W_{t}}{4{\left(\nu_{e}-1\right)}^{2}h_{\max}^{3}}$$

The elastic modulus also can be calculated by substituting Equation (21) into Equation (16). The hardness and elastic modulus are obtained by fitting curve method to process the load–displacement curve obtained in the experiment, as shown in Figure 8. The hardness of the PP/PE/PP separator (specimen 1) and PE separator (specimen 2) is between 34.71 MPa and 40.13 MPa, and 16.10 MPa and 25.13 MPa, respectively. The elastic modulus of the PP/PE/PP separator (specimen 1) and PE separator (specimen 2) is between 0.47 GPa and 0.53 GPa, and 0.51 GPa and 0.71 GPa, respectively. The hardness of the PP/PE/PP separator (specimen 1) is greater than that of the PE separator (specimen 2), while the elastic modulus of the PP/PE/PP separator (specimen 1) is smaller than that of the PE separator (specimen 2) as a whole when the thickness is same.

### 3.5. Comparison among Three Theoretical Methods

The nanoindentation experiment data of specimens are shown in Table A1 of Appendix A. The hardness and elastic modulus of specimens determined by Oliver–Pharr method, indentation work method and fitting curve method are shown in Table A2 of Appendix A. The hardness of specimen obtained by the fitting curve method is very close to that of the specimen obtained by the Oliver–Pharr method, but the hardness of specimen obtained by the indentation work method is almost 50% lower than that of specimen obtained by the Oliver–Pharr method and fitting curve method. The hardness and elastic modulus of the specimen obtained by the Oliver–Pharr method is the highest due to the deviation in the calculation of contact area, while the hardness and elastic modulus of the specimen obtained by the indentation work method is the lowest due to the deviation caused by replacing the contact depth with maximum indentation depth. The hardness and elastic modulus obtained by fitting curve method is close to the average value of the Oliver–Pharr method and indentation work method. Therefore, the stacking effect (“sink-in” phenomenon) exists in the soft separator. The comparison of hardness and elastic modulus determined by the Oliver–Pharr method, indentation work method, and fitting curve method is shown in Figure 9. Due to the fitting exponential, *n* is less than 2 in Table A1, the total work is too large based on the Kick’s law (*P* = *Ch*^2^), which leads to less hardness and elastic modulus. The results obtained by the fitting curve method are more reasonable and accurate, which not only avoids the problem of large contact area obtained by the Oliver–Pharr method, but also avoids the influence caused by the large fitting data of displacement–force curve and the inaccuracy of using the maximum displacement obtained by the indentation work method.

It can be seen from Figure 9 and Table A2 that the changes of the hardness and elastic modulus of specimens determined by Oliver–Pharr method, indentation work method, and fitting curve method are consistent at the different indentation points. The average values at the different indentation points of three methods are shown in Figure 9. Take the specimen determined by fitting curve method for example, the hardness of PP/PE/PP separator is between 34.71 MPa and 40.13 MPa, with the average value of 37.30 MPa and an average deviation of 1.32 MPa. The difference between the maximum and the minimum values is 5.42 MPa, which accounts for 14.5% of the average value, as shown in Figure 9a. The research results show that the hardness of PP/PE/PP separator measured at different indentation points has little difference and the dispersion degree is low. The overall curve is relatively stable without significant change, which indicates that the local displacement of separator that resists the external load pressed into its surface is average because the separator structure is relatively uniform. The micropore of separator has little influence on the obstruction of the electrode active material particles at the initial stage and the micro particles, such as fine metal powder, that may enter the lithium-ion battery during the manufacturing process.

Whereas the hardness of the PE separator measured at different indentation points has great difference and the dispersion degree is high, which ranges from 16.10 MPa and 25.13 MPa with the average value of 21.99 MPa and an average deviation of 2.43 MPa, as shown in Figure 9b. The difference between the maximum and the minimum values is 9.03 MPa, which accounts for 41.1% of the average value. Compared with the hardness of the PP/PE/PP separator, the hardness of the PE separator measured at different indentation points has great difference and the dispersion degree is high. The overall curve changes dramatically, which indicates that the bearing capacity of separator decreases because the local displacement of separator that resists the external load pressed into its surface varies greatly. The micropores of the separator have a significant influence on the obstruction of the electrode active material particles at the initial stage and on the micro particles that may enter the lithium-ion battery during the manufacturing process, which may cause damage to the weak point of the separator.

The change trend of elastic modulus of the PP/PE/PP separator are same as that of the hardness, which changes in a relatively stable manner without significant change. The elastic modulus of the PP/PE/PP separator is between 0.47 GPa and 0.53 GPa with the average value of 0.50 GPa and an average deviation of 0.02 GPa. The difference between the maximum and the minimum values is 0.06 GPa, which accounts for 12% of the average value, as shown in Figure 9c. The research results show that the mechanical properties of the PP/PE/PP separator are barely affected by the micropores in the separator due to the good uniformity of the separator. While the change trend of elastic modulus of PE separator are same as that of the hardness, which changes dramatically with significant fluctuation. The elastic modulus of PE separator is between 0.51 GPa and 0.71 GPa with the average value of 0.62 GPa and an average deviation of 0.05 GPa. The difference between the maximum and the minimum values is 0.20 GPa, which accounts for 32.26% of the average value, as shown in Figure 9d. The research results show that the mechanical properties of PE separator are great affected by the micropores in the separator due to the overall curve changes dramatically.

The hardness of the PP/PE/PP separator is greater than that of the PE separator. The greatest hardness of the PE separator at the Number 6 indentation point is almost only 73.50% of the lowest hardness of the PP/PE/PP separator, as shown in Figure 9a,b. It is indicated that the obstruction ability of the PP/PE/PP separator to locally resist the external load pressed into its surface and to resist micro particles, such as fine metal powder, that can enter the lithium-ion battery during the manufacturing process is greater than that of the PE separator. The elastic modulus of the PE separator is greater than that of the PP/PE/PP separator, as shown in Figure 9c,d. It is indicated that the elastic deformation resistance of the PE separator is stronger than that of the PP/PE/PP separator. The PE separator may undergo recoverable deformation under a certain small stress, while the PP/PE/PP separator has plastic deformation and cannot be recovered. Therefore, it can be concluded that the elasticity of the PE separator is better than that of the PP/PE/PP separator during the shallow compression stage.

## 4. Conclusions

The nanoindentation experiment was performed to investigate the mechanical properties of two types of separators for LIBs based on the grid nanoindentation method. During the indentation experiment, the “sink-in” phenomenon was observed around the indenter when plastic deformation of the specimen occurred. The “sink-in” area of PE separator is larger than that of PP/PE/PP separator, i.e., the plastic area of the PE separator was larger than that of PP/PE/PP separator. In order to select a suitable method to evaluate the hardness and elastic modulus of these separators for LIBs, three theoretical methods, including the Oliver–Pharr method, the indentation work method, and the fitting curve method, were used for analysis and comparison in this study. The results obtained by the fitting curve method are more reasonable and accurate, not only avoiding the problem of large contact area obtained by the Oliver–Pharr method, but also avoiding the influence caused by the large fitting data of the displacement–force curve and the inaccuracy of using the maximum displacement obtained by the indentation work method. In addition, the obstruction ability of the PP/PE/PP separator to locally resist the external load pressed into its surface and to resist the micro particles, such as fine metal powder, that can enter the lithium-ion battery during the manufacturing process is stronger than that of the PE separator. This research provides guidance for studying the mechanical properties and exploring the estimation method of macromolecular separators for LIBs.

## Figures and Tables

**Figure 1 polymers-14-03664-f001:**
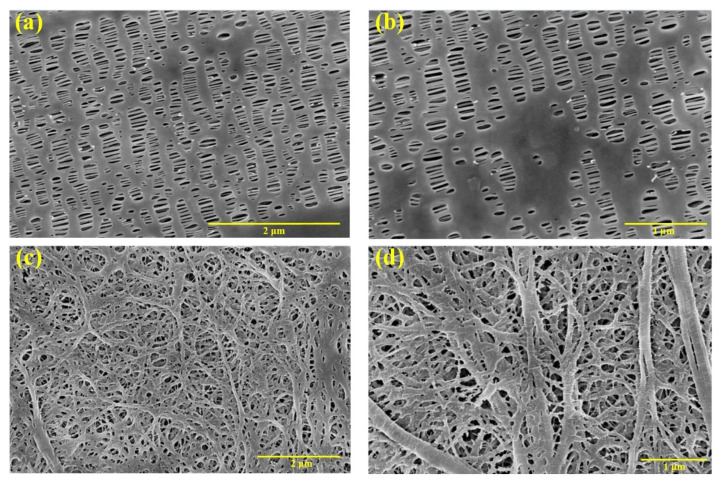
Microstructures of in-plane view of PP/PE/PP and PE separators: (**a**) PP/PE/PP (8 × 10^4^ times); (**b**) PP/PE/PP (1 × 10^5^ times); (**c**) PE (5 × 10^4^ times); (**d**) PE (8 × 10^4^ times).

**Figure 2 polymers-14-03664-f002:**
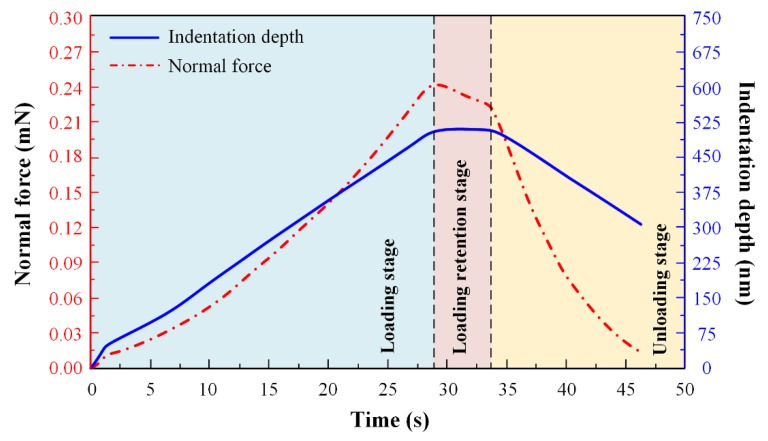
Indentation depth and normal force curves of nanoindentation experiment.

**Figure 3 polymers-14-03664-f003:**
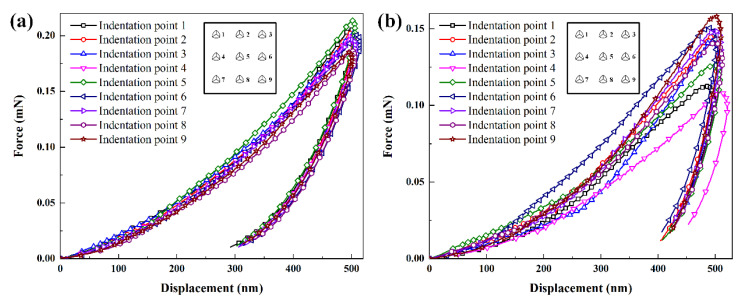
The displacement–force curves of specimens at different indentation points: (**a**) specimen 1; (**b**) specimen 2.

**Figure 4 polymers-14-03664-f004:**
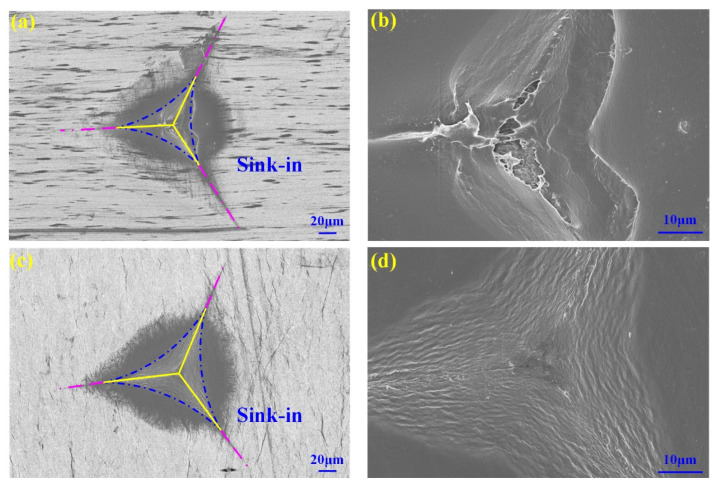
Micromorphology of in-plane view of specimens: (**a**) specimen 1 (300 times); (**b**) specimen 1 (1.5 × 10^3^ times); (**c**) specimen 2 (300 times); (**d**) specimen 2 (1.5 × 10^3^ times).

**Figure 5 polymers-14-03664-f005:**
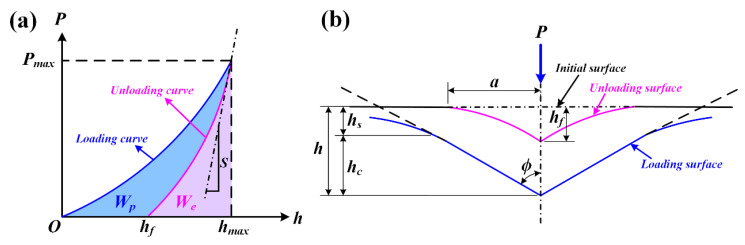
Schematic illustration of theoretical model of the Oliver–Pharr method: (**a**) displacement–force curve showing important measured parameters; (**b**) unloading process showing parameters characterizing the contact geometry.

**Figure 6 polymers-14-03664-f006:**
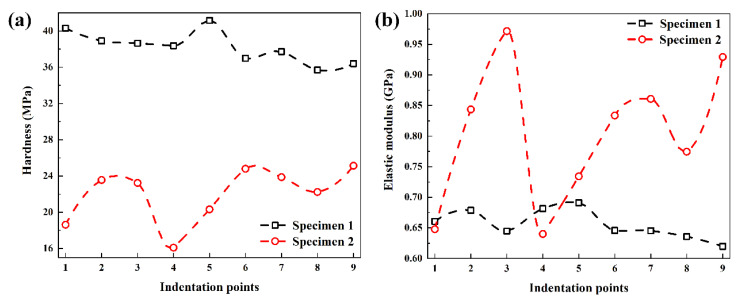
Mechanical parameters of specimens based on the Oliver–Pharr method: (**a**) hardness; (**b**) elastic modulus.

**Figure 7 polymers-14-03664-f007:**
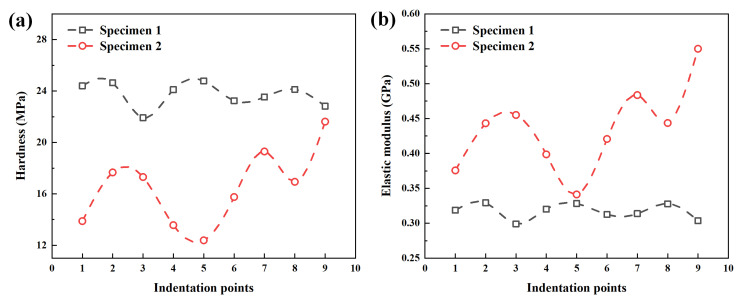
Mechanical parameters of specimens based on the indentation work method: (**a**) hardness; (**b**) elastic modulus.

**Figure 8 polymers-14-03664-f008:**
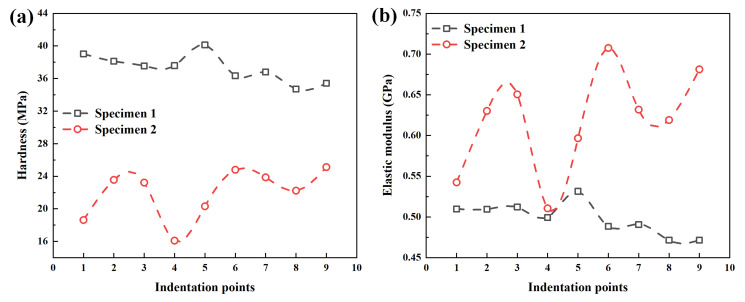
Mechanical parameters of specimens based on the fitting curve method: (**a**) hardness; (**b**) elastic modulus.

**Figure 9 polymers-14-03664-f009:**
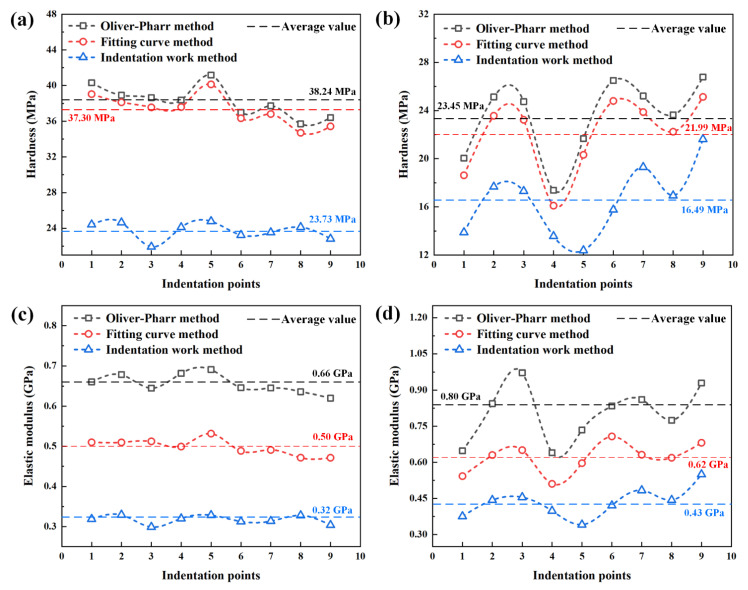
Hardness and elastic modulus of specimens determined by Oliver–Pharr method, indentation work method and fitting curve method: (**a**) hardness of PP/PE/PP separator; (**b**) hardness of PE separator; (**c**) elastic modulus of PP/PE/PP separator; (**d**) elastic modulus of PE separator.

**Table 1 polymers-14-03664-t001:** Fundamental properties of tri-layer polypropylene/polyethylene/polypropylene (PP/PE/PP) and single-layer polyethylene (PE) [30,31,32].

No.	Separators	Thickness (µm)	Permeability (s)	Porosity (%)	Pore Size (µm × µm)	Tensile Strength (MPa)		Puncture Strength (N)	Thermal Stability (Shrinkage) 90 °C/1 h (%)	
						TD	MD		TD	MD
1	Tri-layer (PP/PE/PP)	25	620	39	0.5 × 0.05	14.71	166.71	>3.724	0	<5
2	Single-layer (PE)	25	240	40	0.06 × 0.06	13.73	139.25	>3.054	0	<5

**Table 2 polymers-14-03664-t002:** Control parameters of nanoindentation experiment.

Control Parameters	Data Acquisition Rate (Hz)	Maximum Linear Loading Depth (nm)	Loading Rate (nm/min)	Unloading Rate (nm/min)	Initial Contact Distance (nm)	Indenter Entrance Rate (nm/min)	Indenter Exit Rate (nm/min)	Stiffness Threshold (μN/μm)
Value	10	500	1000	1000	2000	2500	2000	150

## Data Availability

Not applicable.

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
