# Peer review of "Mechanical Properties of Macromolecular Separators for Lithium-Ion Batteries Based on Nanoindentation Experiment"

_polymers, 2022, doi:10.3390/polym14173664_

Round 1
Reviewer 1 Report
In this study Tingting Xu and coworkers report on the mechanical properties of hardness and elasticity for separators typically applied in Lithium-ion battery systems. The paper reports on 3 different strategies to determine these properties, namely the Oliver-Pharr method, the indentation work method, and the fitting curve method. The reason that the paper reports applying these varying methods is due to the error that arises from the “sink-in” phenomenon during the collection of the indentation curve data and how these ranges in data can result in differing results during the various fitting methods. The elastic and plastic properties of the material polyethylene (PE) and polypropylene/polyethylene/polypropylene (PP/PE/PP) have been analyzed to then determine the hardness and elasticity of the samples.
Key Issues:
l Despite the paper focusing on nanoindentation methods applied to LIB separators, little mention of the required properties of an LIB separator are made aside from the very basics. The paper would be more impactful if the authors could convey to the reader what levels hardness and elasticity are optimal for LIB application and why. Mentioning the requirements of different battery system types may also be of use, for example a Lithium metal battery may require a separator with an increased resistance to dendrite puncture. Another example could be the discussion of pouch cell vs cylindrical cell, where the volume expansion and pressure in the system can be drastically different. Including this type of battery information will make the paper more citable by its target audience which is assumably battery focused.
l The introduction mentions that the paper is addressing the safety issue of separators for LIBs which it does not, as no safety recommendations are made; choose wording carefully. The separator failure mechanisms in batteries should also be described so that the reader can understand the impact of characterizing the data and why overshooting or undershooting in the analysis may result in safety issues.
l The original displacement-force curves do not come with enough explanation or emphasis on their importance to the overall findings. Analyzing these curves, as we can find Work and elastic response from them, will help to clarify why the data at the end is as it is. Linking these results mentioned in Figure 3 and 4 to the final data will help to confirm the fittings.
Minor Issues (Authors Discretion):
l Maintain consistency in Analysis such as SEM Mag.
l When considering figures, if the text says to refer to the figure for certain data, even though these are evident for the author it would be clearer to highlight them. (Example Figure 2 “stages”).
l The data in Fig. 3 and 4 seems related and combining them could make it easier for the reader to digest.
l The average values are discussed for each fitting technique, could these be represented in the figures in some way?
l Some tables with large amounts of data are unsightly and could be moved to supplementary.
The links discussed in the key issues section should be addressed before publication.
Author Response
The point-by-point response to the reviewer’s comments is uploaded as a Word file.

Reviewer 2 Report
Comments to authors:
In this manuscript, the authors compared the mechanical strength such as the hardness and elastic modulus of two types of separators for lithium-ion batteries. The authors also combined theoretical methods to analyze the mechanical property of separators. The content of this work is comprehensive and informative. I recommend this work to be published on Polymers after the following revision. Here are some comments.
1. The property of separators might be changed in lithium-ion batteries with the presence of liquid electrolyte. The authors are recommended to compare the mechanical property of these three separators after immersed in liquid electrolyte.
2. The separators should work well in LIBs at various temperatures (for example from 0 to 60 oC). Have the authors studied the effect of temperature on the mechanical properties of separators?
3. How about the mechanical properties of single PP separator?
4. These polymer separators are non-conductive during SEM measurement. The authors are recommended to provide detailed information about how to prepare the SEM sample.
5. In Figure 1, the authors should provide a clear scale bar in each image.
6. In Figure 2, the authors should provide the meaning of X axis.
7. Please provide the vendor of separators used in this work.
Author Response

(The authors gave the same response as above.)

Round 2
Reviewer 2 Report
Accept in present form